# Controlled and Synchronised Vascular Regeneration upon the Implantation of Iloprost- and Cationic Amphiphilic Drugs-Conjugated Tissue-Engineered Vascular Grafts into the Ovine Carotid Artery: A Proteomics-Empowered Study

**DOI:** 10.3390/polym14235149

**Published:** 2022-11-26

**Authors:** Larisa Antonova, Anton Kutikhin, Viktoriia Sevostianova, Arseniy Lobov, Egor Repkin, Evgenia Krivkina, Elena Velikanova, Andrey Mironov, Rinat Mukhamadiyarov, Evgenia Senokosova, Mariam Khanova, Daria Shishkova, Victoria Markova, Leonid Barbarash

**Affiliations:** 1Department of Experimental Medicine, Research Institute for Complex Issues of Cardiovascular Diseases, 6 Sosnovy Boulevard, Kemerovo 650002, Russia; 2Department of Regenerative Biomedicine, Research Institute of Cytology, 4 Tikhoretskiy Prospekt, Saint Petersburg 194064, Russia; 3Centre for Molecular and Cell Technologies, Saint Petersburg State University, Universitetskaya Embankment, 7/9, Saint Petersburg 199034, Russia

**Keywords:** vascular tissue engineering, tissue-engineered vascular grafts, poly(ε-caprolactone), iloprost, cationic amphiphilic drugs, proteomic profiling, vascular smooth muscle cells, basement membrane, vascular regeneration, chronic inflammation

## Abstract

Implementation of small-diameter tissue-engineered vascular grafts (TEVGs) into clinical practice is still delayed due to the frequent complications, including thrombosis, aneurysms, neointimal hyperplasia, calcification, atherosclerosis, and infection. Here, we conjugated a vasodilator/platelet inhibitor, iloprost, and an antimicrobial cationic amphiphilic drug, 1,5-bis-(4-tetradecyl-1,4-diazoniabicyclo [2.2.2]octan-1-yl) pentane tetrabromide, to the luminal surface of electrospun poly(ε-caprolactone) (PCL) TEVGs for preventing thrombosis and infection, additionally enveloped such TEVGs into the PCL sheath to preclude aneurysms, and implanted PCL^Ilo/CAD^ TEVGs into the ovine carotid artery (n = 12) for 6 months. The primary patency was 50% (6/12 animals). TEVGs were completely replaced with the vascular tissue, free from aneurysms, calcification, atherosclerosis and infection, completely endothelialised, and had clearly distinguishable medial and adventitial layers. Comparative proteomic profiling of TEVGs and contralateral carotid arteries found that TEVGs lacked contractile vascular smooth muscle cell markers, basement membrane components, and proteins mediating antioxidant defense, concurrently showing the protein signatures of upregulated protein synthesis, folding and assembly, enhanced energy metabolism, and macrophage-driven inflammation. Collectively, these results suggested a synchronised replacement of PCL with a newly formed vascular tissue but insufficient compliance of PCL^Ilo/CAD^ TEVGs, demanding their testing in the muscular artery position or stimulation of vascular smooth muscle cell specification after the implantation.

## 1. Introduction

In spite of significant efforts on developing biodegradable, durable, haemo- and biocompatible, and triple-resistant (i.e., calcification-, infection-, and atheroresistant) small-diameter tissue-engineered vascular graft (TEVG), with compliance and elastance similar to the native arteries, current advances in this field are limited to the experimental prototypes showing at best 50% primary patency in the long term (e.g., 1 year postimplantation) [1,2,3,4]. Therefore, TEVGs still represent an unmet clinical need, despite more than two decades of development [1,2]. Albeit, a number of natural and synthetic polymers, as well as their combinations, possess high biocompatibility and elasticity, drawbacks limiting the clinical translation of existing TEVGs still include insufficient haemocompatibility, biomechanical incompetence, neointimal hyperplasia, atherosclerosis, and calcification [5,6,7,8]. Together, these shortcomings invalidate the advantages of TEVGs, i.e., complete replacement with the regenerating vascular tissue that ensures the adaptation to body growth (particularly important in pediatric surgery [9,10]) and no need to harvest autologous blood vessels from other sites (beneficial for the bypass surgery [11,12]). In a TEVG implantation setting, vascular regeneration is provided by a biosimilar structure of the electrospun scaffold (micro- to nanoscale fiber networks which form multiple interconnected pores), rapid recruitment and proliferation of progenitor cells and their vascular differentiation guided by shear stress, cyclic pressure, and circulating growth factors [13,14,15].

The most frequently used biodegradable polymer in vascular tissue engineering is poly(ε-caprolactone) (PCL), a synthetic compound which is sufficiently durable and elastic to withstand blood pressure until replacement of PCL with de novo vascular tissue [16,17]. Although biocompatibility of PCL is lower than in natural polymers, it does not provoke inadequate immune response and shows a stable biodegradation pattern [16,17]. Good processability during electrospinning and relatively low cost highlight PCL as a promising polymer for the mass production of TEVGs [16,17], and tubular PCL scaffolds can be efficiently tailored with other polymers such as fibrin [18], chitosan [19], and collagen [20] to improve implantation results.

Unmodified PCL vascular grafts show insufficient haemocompatibility upon the implantation into sheep, a well-accepted “worst case model” in vascular tissue engineering which mimics: (1) pro-thrombotic conditions in patients with hereditary thrombophilia, cancer, autoimmune disorders, nephrotic and metabolic syndromes, obesity, prolonged immobilisation, and regular use of oral contraceptives; (2) pro-calcification conditions, i.e., advanced age, chronic kidney disease, diabetes mellitus, and osteopoenia/osteoporosis; (3) pro-aneurysmal conditions because of high blood velocity in elastic carotid artery (which is a common site of TEVG implantation in sheep) as compared to muscular arteries including coronary and internal thoracic arteries [21,22]. Clots within the scaffold promote microbial adhesion in cases of transient bacteriemia similar to the infective endocarditis scenario [23], and vascular graft infection might further enhance thrombosis [24]. Previously, we attached an anticoagulant, heparin, and a vasodilator/antiplatelet drug, iloprost, to the luminal surface of PCL-based TEVGs and reached 50% patency in the ovine model 18 months postimplantation, also demonstrating complete vascular regeneration at the implantation site [25]. Other groups have also reported efficiency of heparin conjugation for preventing thrombotic complications upon the TEVG implantation into sheep [26,27] and swine [28]. However, such coating lacks antimicrobial activity.

Another obstacle in the field is the dearth of unbiased mass spectrometry profiling in studies investigating vascular regeneration during TEVG degradation and synchronisation of these processes [29], as proteomic analysis has been limited to decellularised human umbilical [30,31] or placental [32] arteries. As compared to the proteomics, an RNA sequencing approach is inferior for TEVG analysis, as differentially expressed transcripts are often not translated correspondingly and it does not reflect the molecules which emerge from the systemic circulation rather than being produced in situ.

Here, we showed that biodegradable tissue-engineered vascular grafts (TEVGs) modified with a vasodilator/platelet inhibitor, iloprost, and antimicrobial cationic amphiphilic drugs have non-inferior 6-month primary patency (50%) upon the implantation into the ovine carotid artery. To uncover the molecular events behind the TEVG remodeling, we, for the first time, performed label-free proteomic profiling of TEVGs and found them free of enzymatic degradation, thrombosis, aneurysms, calcification, atherosclerosis, or infection. Further, TEVGs demonstrated a complete endothelialization but lacked the basement membrane and hierarchical structure of the medial layer. In spite of a strong macrophage-driven inflammation, TEVGs fully retained their integrity, indicative of curbed proteolysis and controlled remodeling. Hence, we suggested that conjugation of cationic amphiphilic drugs and iloprost to the luminal surface endows TEVGs with perfect haemocompatibility and contributes to the physiological replacement of PCL with newly formed vascular tissue which, however, lacks regular elastic laminae and vascular smooth muscle cell layers, thereby posing insufficient compliance.

## 2. Materials and Methods

### 2.1. Fabrication of TEVGs

Biodegradable TEVGs (4 mm diameter, ≈ 400 μm thickness and 4 cm length) were fabricated using electrospinning (Nanon-01A, MECC, Tokyo, Japan) from 12% PCL (440744, Sigma-Aldrich, Saint Louis, MO, USA) dissolved in 1,1,1,3,3,3-hexafluoro-2-propanol (366927, Sigma-Aldrich, Saint Louis, MO, USA) solution. Electrospinning parameters were as follows: 22 kV voltage, 22G needle, 0.5 mL/h feed rate, 4 mm rotating drum diameter, 1000 rpm collector rotation speed, and 150 mm tip-to-collector distance. A metal pin with a 4 mm diameter was used as a collector. Upon the fabrication, TEVGs were coated with anti-aneurysmatic PCL (440744, Sigma-Aldrich, Saint Louis, MO, USA) sheath by fused deposition modeling using an original 3D printer extruder and the following extrusion parameters: 1 rps roller speed, 1 mm/sec carriage speed, 0.5 mm extruder diameter, 0.5 mm/sec feed rate, and 160 °C feed temperature.

### 2.2. Anti-Thrombotic and Antimicrobial Modification and Sterilisation of TEVGs

To increase haemocompatibility of TEVGs, we first immobilised a poly(vinylpyrrolidone) (PVP) hydrogel at their luminal surface by incubating TEVGs in 5% alcoholic solution of PVP (K90, PanReac AppliChem, Darmstadt, Germany) for 30 min, drying at room temperature and sterile conditions for 24 h, and irradiating in the argon atmosphere by a linear particle accelerator (ILU-10, 15 kGy, electron energy 5 mEV, beam power 50 kW, Budker Institute of Nuclear Physics of Siberian Branch of the Russian Academy of Sciences, Novosibirsk, Russian Federation). As irradiation also sterilised TEVGs, subsequent procedures were conducted under sterile conditions. Then, TEVGs were cleansed from unconjugated polymer by washing in sterile water for 60 min. PVP conjugation was verified by Fourier-transform infrared spectroscopy (FTIR, Vertex 80v, Bruker, Billerica, MA, USA) at a resolution of 4 cm^−1^ and at wavelengths ranging from 4000 to 500 cm^−1^ (FTIR).

Anti-thrombotic coating was created using iloprost (Bayer, Leverkusen, Germany) whilst antimicrobial effect was furnished with a cationic amphiphilic drug (CAD) 1,5-bis-(4-tetradecyl-1,4-diazoniabicyclo [2.2.2] octan-1-yl) pentane tetrabromide (NanoTech-S, Novosibirsk, Russia). The synthesis of the latter was conducted as previously described [33]. Coating was performed by incubating PVP-coated TEVGs with iloprost (0.2 µg/mL) and 1,5-bis-(4-tetradecyl-1,4-diazoniabicyclo [2.2.2] octan-1-yl) pentane tetrabromide (0.25 mg/mL dissolved in methanol) for 30 min with the following drying at room temperature under sterile conditions.

### 2.3. Conjugation Verification and Controlled Release Evaluation

For both attached drugs (i.e., iloprost and 1,5-bis-(4-tetradecyl-1,4-diazoniabicyclo [2.2.2] octan-1-yl) pentane tetrabromide), we assessed their conjugation with TEVGs and release kinetics. To investigate the controlled release of iloprost and 1,5-bis-(4-tetradecyl-1,4-diazoniabicyclo [2.2.2] octan-1-yl) pentane tetrabromide, TEVGs were incubated in the sterile phosphate-buffered saline (PBS, P071, Pushchino Laboratories, Pushchino, Russia) at 37 °C, and PBS was sequentially collected at 1, 5, 10, 15, 20, 30, 45, 60, 75 and 90 days.

Semi-quantitative detection of iloprost on TEVG luminal surface and time-resolved evaluation of its release into the milieu at all indicated time points was performed by Fourier-transform infrared spectroscopy (FTIR, Infralum FT-08, Lumex Instruments, Saint Peterburg, Russia) employing a zinc selenide attenuated total reflectance accessory (Pike Technologies, Fitchburg, WI, USA). Around 1 cm^2^ piece of each TEVG luminal surface (or phosphate-buffered saline co-incubated with TEVGs) was analysed, and the spectra were obtained over the 620–4000 cm^−1^ region at a resolution of 4 cm^−1^ with a total of 64 scans. For the semi-quantitative analysis, the reflectance spectra (R) were converted to Kubelka–Munk equivalent absorbance units (F) according to the Equation (1), as the intensities of the absorption bands of many substances are proportional to their concentrations in the Kubelka–Munk model: F(R) = (1 − R)^2^/2R(1)

Bromine, an element appearing in TEVGs specifically upon the attachment of CAD 1,5-bis-(4-tetradecyl-1,4-diazoniabicyclo [2.2.2] octan-1-yl) pentane tetrabromide, was determined by energy-dispersive X-ray spectroscopy immediately after the fabrication and after storage at 4 °C for 1 day, at −20 °C for 1 day or 6 months, or after a 6 h incubation in PBS supplied with 0.05% Tween-20 at 37 °C. For energy-dispersive X-ray spectroscopy, dried samples were mounted on an aluminum stub using double-sided adhesive carbon tape, and sputter coated with carbon (60 nm thickness) using EM ACE 200 sputter coater (Leica Microsystems, Wetzlar, Germany). Energy-dispersive X-ray spectroscopy was conducted using a scanning electron microscope S-3400N (Hitachi, Tokyo, Japan) coupled with a high-resolution silicon drift detector (XFlash 4010, Bruker, Billerica, MA, USA) in a secondary electron mode at 20 kV accelerating voltage and high vacuum.

The release profile of 1,5-bis-(4-tetradecyl-1,4-diazoniabicyclo [2.2.2] octan-1-yl) pentane tetrabromide into the milieu (PBS) was assessed by liquid chromatography–mass spectrometry (LC-MS, Agilent 1100 Series Liquid Chromatograph/Mass Selective Detector, Agilent Technologies, Santa Clara, CA, USA) at 5 and 90 days.

### 2.4. Implantation of TEVGs into the Ovine Carotid Artery

The study was conducted according to the guidelines of the Declaration of Helsinki, and was approved by the Local Ethical Committee of the Research Institute for Complex Issues of Cardiovascular Diseases (Kemerovo, Russian Federation, protocol code 2020/06, date of approval: 19 February 2020). Animal experiments were performed in accordance with the European Convention for the Protection of Vertebrate Animals (Strasbourg, 1986) and Directive 2010/63/EU of the European Parliament on the protection of animals used for scientific purposes. For the implantation, we used female Edilbay sheep of 42–45 kg body weight which were received from the Animal Core Facility of the Research Institute for Complex Issues of Cardiovascular Diseases (Kemerovo, Russian Federation) and selected for the surgery by Doppler ultrasonography to identify those having carotid artery diameter of 4.0 ± 0.2 mm. Biodegradable PCL^Ilo/CAD^ TEVGs (n = 12, one graft per animal) have been implanted for 6 months.

Upon the premedication with xylazine (0.005–0.025 mL/kg body weight intramuscularly, Nita-Pharm, Saratov, Russia) and atropine (1 mg intramuscularly, Moscow Endocrine Plant, Russia) and induction into the anaesthesia with propofol (5–7 mg per kg body weight intravenously, Armavir Bioplant, Armavir, Russia) and atracurium besylate (0.5–0.6 mg per kg body weight intravenously, Biokhimik, Saransk, Russia), it was maintained by the endotracheal intubation and artificial ventilation with sevoflurane (2–4% (*v*/*v*), R-Pharm, Moscow, Russia) and intravenous infusion of atracurium besylate (0.3–0.6 mg/kg/h, Biokhimik, Saransk, Russia) with the constant control of heart rate, respiratory rate, blood pressure, and oxygen saturation. Through the whole time of the surgery, artificial ventilation parameters were as follows: 12–15/min respiratory rate, 7–9 mbar positive end-expiratory pressure (PEEP), 6–8 mL/kg respiratory volume, and 40–60% fraction of inspired oxygen (FiO_2_).

Following the access to the carotid artery and intravenous infusion of heparin (5000 IU, Velpharm, Moscow, Russia), we clamped the artery, excised a 4 cm segment, performed end-to-end implantation of a TEVG using the twisted seam (Prolene 6-0, Ethicon, Somerville, NJ, USA), and conducted the wound closure (Vicryl 2-0, Ethicon, Somerville, NJ, USA). During the surgery, animals received 0.9% NaCl (500 mL intravenously, Hematek, Tver, Russia) and cefuroxime (1.5 g intravenously, Biokhimik, Saransk, Russia). After the wound closure, we injected an inhibitor of factor Xa enoxaparin sodium (4000 IU/0.4 mL subcutaneously, Sotex, Moscow, Russia) and performed the extubation. Postoperative care included injections of cefuroxime (1.5 g intramuscularly twice daily, Biokhimik, Saransk, Russia), enoxaparin sodium (4000 IU/0.4 mL subcutaneously daily, Sotex, Moscow, Russia), clopidogrel (75 mg orally daily, Kanonpharma Production, Moscow, Russia), and heparin sodium (5000 IU subcutaneously twice daily, Velpharm, Moscow, Russia) during 5 days. Graft patency was assessed by Doppler ultrasonography immediately after the surgery and then at the following time points: 1 day, 3 days, 1 month, 3 months, and 6 months postoperation. At 6 months postoperation, sheep were sacrificed. Excised TEVGs and contralateral healthy carotid arteries were cut into four segments of equal length. The proximal segment was selected for the histological examination, two midgraft segments were used for the immunostaining and electron microscopy, and the distal segment was employed for the proteomic analysis.

### 2.5. Histological Examination

The proximal segment was fixed in two changes of 10% neutral phosphate buffered formalin (B06-003, ErgoProduction, Saint Petersburg, Russia) for 24 h at 4 °C, embedded into paraffin (Histomix Extra, 10342, ErgoProduction, Saint Petersburg, Russia), and cut (8 µm sections) on a microtome (Microm HM 325, Thermo Scientific, Waltham, MA, USA) as previously described [25]. To ensure proper histological examination, we prepared 8 sections, evenly distributed across the entire excised segment, per slide. Upon the deparaffinisation in three changes of xylene (X0053, Diapath, Martinengo, Italy) and three changes of 95% ethanol (Kemerovo Pharmaceutical Plant, Kemerovo, Russia), sections were stained with: (1) haematoxylin and eosin (05-003 and 05-011, ErgoProduction, Saint Petersburg, Russia) as described in [34] for the general examination; (2) van Gieson stain (21-020, ErgoProduction, Saint Petersburg, Russian Federation) as described in [34] to distinguish connective and smooth muscle tissue; (3) 2 % aqueous alizarin red S (6-09-1749-77, Reachim, Moscow, Russia) and 4′,6-diamidino-2-phenylindole (DAPI, 10 μg/mL, D9542, Sigma-Aldrich, Saint Louis, MO, USA) as described in [34] for the detection of calcium deposits within the grafts. Visualisation was performed by light or fluorescent microscopy (AxioImager.A1, Carl Zeiss, Oberkochen, Germany).

### 2.6. Immunostaining

The first midgraft segment was snap-frozen in the optimal cutting temperature medium (Tissue-Tek, 4583, Sakura Finetek, Tokyo, Japan) and cut on a cryostat (Microm HM 525, Thermo Scientific, Waltham, MA, USA) as described above. Before the immunostaining, sections (8 µm thickness) were fixed in 4% paraformaldehyde (158127, Sigma-Aldrich, Saint Louis, MO, USA) for 10 min, permeabilised in Triton X-100 (T8787, Sigma-Aldrich, Saint Louis, MO, USA) for 15 min, and blocked in 1% bovine serum albumin (P091E, PanEco, Moscow, Russia) for 1 h to prevent non-specific binding. Sections were then stained with rabbit anti-CD31 (1:200 dilution, ab28364, Abcam, Cambridge, UK) and mouse anti-α-SMA (1:200, ab7817, Abcam, Cambridge, UK), rabbit anti-vWF (1:200, ab6994, Abcam, Cambridge, UK), rabbit anti-collagen IV (1:200, ab6586, Abcam, Cambridge, UK), rabbit anti-collagen III (1:200, NB600-594, Novus Biologicals, Centennial, CO, USA), rabbit anti-CD163 (1:200, ab182422, Abcam, Cambridge, UK), or mouse anti-CD68 (1:200, M0718, Agilent Technologies, CA, USA) primary antibodies overnight at 4 °C. Next day, sections were further treated with goat anti-rabbit highly cross-adsorbed Alexa Fluor 488-conjugated (1:500, A11034, Thermo Fisher Scientific, Waltham, MA, USA) and donkey anti-mouse highly cross-adsorbed Alexa Fluor 555-conjugated (1:500, A31570, Thermo Fisher Scientific, Waltham, MA, USA) secondary antibodies for 1 h at room temperature. Autofluorescence was eliminated with Autofluorescence Eliminator Reagent (2160, Merck Millipore, Burlington, MA, USA) according to the manufacturer’s protocol. Counterstaining was performed with DAPI (10 μg/mL, D9542, Sigma-Aldrich, Saint Louis, MO, USA) for 30 min. At all stages, washing was conducted with 0.1% saline (60201, Pushchino Laboratories, Pushchino, Russia) solution of Tween-20 (P9416, Sigma-Aldrich, Saint Louis, MO, USA). Coverslips were mounted with ProLong Gold Antifade (P36934, Thermo Fisher Scientific, Waltham, MA, USA). Visualisation was performed by a confocal microscopy (LSM700, Carl Zeiss, Oberkochen, Germany).

### 2.7. Ultrastructural Analysis

After the electrospinning, TEVGs were sputter coated with gold and palladium (Leica EM ACE200, Leica Microsystems, Wetzlar, Germany) and visualised by scanning electron microscopy (S-3400N, Hitachi, Tokyo, Japan).

The second midgraft segment was fixed in 10% neutral phosphate buffered formalin (B06-003, ErgoProduction, Saint Petersburg, Russia), postfixed and stained in 1% phosphate buffered osmium tetroxide (OsO_4_, 19110, Electron Microscopy Sciences, Hatfield, PA, USA), dehydrated in ascending ethanol series (Kemerovo Pharmaceutical Plant, Kemerovo, Russia) and isopropanol (06-002, ErgoProduction, Saint Petersburg, Russia), stained in 2% alcoholic uranyl acetate (22400-2, Electron Microscopy Sciences, Hatfield, PA, USA), impregnated with acetone (6-09-20-03-83, EKOS-1, Moscow, Russia): epoxy resin (1:1) and epoxy resin (Araldite 502, 13900, Electron Microscopy Sciences, Hatfield, PA, USA), embedded into epoxy resin, grinded, polished, counterstained with Reynolds’s lead citrate (17810, Electron Microscopy Sciences, Hatfield, PA, USA), sputter coated with carbon (Leica EM ACE200, Leica Microsystems, Wetzlar, Germany) and visualised by means of backscattered scanning electron microscopy (S-3400N, Hitachi, Tokyo, Japan) in accordance with an EM-BSEM (embedding and backscattered scanning electron microscopy) procedure as previously described [25,35].

### 2.8. Proteomic Profiling

Distal segments of TEVGs and intact contralateral carotid arteries were flushed with a physiological saline (Hematek, Tver, Russia) and homogenised (FastPrep-24 5G, MP Biomedicals, San Diego, CA, USA; Lysing Matrix S Tubes, 116925050-CF, MP Biomedicals, San Diego, CA, USA) in T-PER buffer (78510, Thermo Fisher Scientific, Waltham, MA, USA) supplied with Halt protease and phosphatase inhibitor cocktail (78444, Thermo Fisher Scientific, Waltham, MA, USA) according to the manufacturer’s protocol. Upon the initial centrifugation at 14,000× *g* (Microfuge 20R, Beckman Coulter, Brea, CA, USA) for 10 min, supernatant was additionally centrifuged at 200,000× *g* (Optima MAX-XP, Beckman Coulter, Brea, CA, USA) for 1 h to sediment insoluble ECM proteins. Quantification of total protein was conducted using BCA Protein Assay Kit (23227, Thermo Fisher Scientific, Waltham, MA, USA) and Multiskan Sky microplate spectrophotometer (Thermo Fisher Scientific, Waltham, MA, USA) in accordance with the manufacturer’s protocol.

Upon the removal of RIPA buffer by acetone precipitation (650501, Sigma-Aldrich, Saint Louis, MO, USA), protein pellet was resuspended in 8 mol/L urea (U5128, Sigma-Aldrich, Saint Louis, MO, USA) diluted in 50 mmol/L ammonium bicarbonate (09830, Sigma-Aldrich, Saint Louis, MO, USA). The protein concentration was measured by Qubit 4 fluorometer (Q33238, Thermo Fisher Scientific, Waltham, MA, USA) with QuDye Protein Quantification Kit (25102, Lumiprobe, Cockeysville, MD, USA) according to the manufacturer’s protocol. Protein samples (15 μg) were then incubated in 5 mmol/L dithiothreitol (D0632, Sigma-Aldrich, Saint Louis, MO, USA) for 1 h at 37 °C with the subsequent incubation in 15 mmol/L iodoacetamide for 30 min in the dark at room temperature (I1149, Sigma-Aldrich, Saint Louis, MO, USA). Next, the samples were diluted with 7 volumes of 50 mmol/L ammonium bicarbonate and incubated for 16 h at 37 °C with 200 ng of trypsin (1:50 trypsin:protein ratio; VA9000, Promega, Madison, WI, USA). The peptides were then frozen at −80 °C for 1 h and desalted with stage tips (Tips-RPS-M.T2.200.96, Affinisep, Le Houlme, France) according to the manufacturer’s protocol using methanol (1880092500, Sigma-Aldrich, Saint Louis, MO, USA), acetonitrile (1000291000, Sigma-Aldrich, Saint Louis, MO, USA), and 0.1% formic acid (33015, Sigma-Aldrich, Saint Louis, MO, USA). Desalted peptides were dried in centrifuge concentrator (Concentrator plus, Eppendorf, Hamburg, Germany) for 3 h and finally dissolved in 20 μL 0.1% formic acid for the further shotgun proteomics analysis.

Approximately 500 ng of peptides were used for shotgun proteomics analysis by ultra-high performance liquid chromatography–tandem mass spectrometry (UHPLC-MS/MS) with ion mobility in a TimsToF Pro mass spectrometer with a nanoElute UHPLC system (Bruker, Billerica, MA, USA). UHPLC was performed in the two-column separation mode with an Acclaim PepMap 5 mm Trap Cartridge (Thermo Fisher Scientific) and a Bruker Fifteen separation column (C18 ReproSil AQ, 150 mm × 0.75 mm, 1.9 µm, 120 A; Bruker, Billerica, MA, USA) in a gradient mode with 400 nL/min flow rate and 40 °C. Phase A was water/0.1% formic acid, phase B was acetonitrile/0.1% formic acid (1000291000, Sigma-Aldrich). The gradient was from 2% to 30% phase B for 42 min, then to 95% phase B for 6 min with subsequent washing with 95% phase B for 6 min. Before each sample, the trap and separation columns were equilibrated with 10 and 4 column volumes, respectively. The CaptiveSpray ion source was used for electrospray ionization with 1600 V of capillary voltage, 3 L/min N2 flow, and 180 °C source temperature. The mass spectrometry acquisition was performed in DDA-PASEF mode with 0.5 s cycle in positive polarity with the fragmentation of ions with at least two charges in m/z range from 100 to 1700 and ion mobility range from 0.85 to 1.30 1/K0.

Protein identification was performed in PEAKS Studio Xpro software (a license granted to Saint Petersburg State University; Bioinformatics Solutions Inc., Waterloo, ON, Canada) using sheep (*Ovis aries*) reference proteome UP000002356 (uploaded on 6 August 2022; 22,110 sequences) and protein contaminants database CRAP (version of 4 March 2019). The search parameters were: parent mass error tolerance 10 ppm and fragment mass error tolerance 0.05 ppm, protein and peptide FDR < 1% and 0.1%, respectively, two possible missed cleavage sites, proteins with ≥ 2 unique peptides. Cysteine carbamidomethylation was set as a fixed modification. Methionine oxidation, N-terminal acetylation, asparagine and glutamine deamidation were set as variable modifications.

The mass spectrometry proteomics data have been deposited to the ProteomeXchange Consortium via the PRIDE [36] partner repository with the dataset identifier PXD036520. Label-free quantification by peak area under the curve and spectral counts was used for the further analysis in R (version 3.6.1; R Core Team, 2019). All proteins presented in ≥ 9 of 12 biological replicates were identified and the groups were compared by “VennDiagram” package [37] and drawing of a Venn diagram. The proteins with NA in >3 of samples in both groups were removed and imputation of missed values by k-nearest neighbors was performed by the “impute” package [38]. Then, log-transformation and quantile normalization with further analysis of differential expression by the “limma” package [39] were conducted. Finally, we carried out clusterisation of samples by sparse partial least squares discriminant analysis in the package “MixOmics” [40]. The “ggplot2” [41] and “EnhancedVolcano” [42] packages were used for visualization. Reproducible code for data analysis is available from https://github.com/ArseniyLobov/Proteomic-profiling-of-grafts-implanted-into-the-ovine-artery (accessed on 03 October 2022).

For the bioinformatic analysis, unique proteins were defined as those presented in ≥9 samples in each group (TEVGs or contralateral healthy carotid arteries) and ≤3 samples of another group. Differentially expressed proteins were defined as those with ≥4-fold higher or lower average peak area as compared to another experimental group as well as those with logarithmical fold change ≥ 1 and false discovery rate-corrected *p* value ≤ 0.05. Bioinformatics analysis was performed using Gene Ontology [43,44], COMPARTMENTS [45], UniProtKB Keywords [46], Reactome [47,48], and Kyoto Encyclopedia of Genes and Genomes (KEGG) [49,50] databases.

### 2.9. Western Blotting

Equal amounts of protein (25 μg per sample) were mixed with NuPAGE lithium dodecyl sulfate sample buffer (NP0007, Thermo Fisher Scientific, Waltham, MA, USA) at a 4:1 ratio and NuPAGE sample reducing agent (NP0009, Thermo Fisher Scientific, Waltham, MA, USA) at a 10:1 ratio, denatured at 99 °C for 5 min, and then loaded on a 1.5 mm NuPAGE 4–12% Bis-Tris protein gel (NP0335BOX, Thermo Fisher Scientific, Waltham, MA, USA). The 1:1 mixture of Novex Sharp pre-stained protein standard (LC5800, Thermo Fisher Scientific, Waltham, MA, USA) and MagicMark XP Western protein standard (LC5602, Thermo Fisher Scientific, Waltham, MA, USA) was loaded as a molecular weight marker. Proteins were separated by the sodium dodecyl sulphate-polyacrylamide gel electrophoresis (SDS-PAGE) at 150 V for 2 h using NuPAGE 2-(N-morpholino)ethanesulfonic acid SDS running buffer (NP0002, Thermo Fisher Scientific, Waltham, MA, USA), NuPAGE Antioxidant (NP0005, Thermo Fisher Scientific, Waltham, MA, USA), and XCell SureLock Mini-Cell vertical mini-protein gel electrophoresis system (EI0001, Thermo Fisher Scientific, Waltham, MA, USA). Protein transfer was performed using polyvinylidene difluoride (PVDF) transfer stacks (IB24001, Invitrogen) and iBlot 2 Gel Transfer Device (Invitrogen) according to the manufacturer’s protocols using a standard transfer mode for 30–150 kDa proteins (P0—20 V for 1 min, 23 V for 4 min, and 25 V for 2 min). PVDF membranes were then incubated in iBind Flex Solution (SLF2020, Solution Kit Thermo Fisher Scientific, Waltham, MA, USA) for 1 h to prevent non-specific binding.

Blots were probed with mouse antibodies to CD68 (1:500, M0718, Agilent Technologies, CA, USA) and glyceraldehyde 3-phosphate dehydrogenase (GAPDH, 1:500, SLM33033M, 1:500, Sunlong Biotech, Hangzhou, Zhejiang, China). Horseradish peroxidase-conjugated goat anti-mouse (AP130P, Sigma-Aldrich, Saint Louis, MO, USA) secondary antibody was used at 1:1000 dilution. Incubation with the antibodies was performed using iBind Flex Solution Kit (SLF2020, Thermo Fisher Scientific, Waltham, MA, USA), iBind Flex Cards (SLF2010, Thermo Fisher Scientific, Waltham, MA, USA) and iBind Flex Western Device (SLF2000, Thermo Fisher Scientific, Waltham, MA, USA) during 3 h according to the manufacturer’s protocols. Chemiluminescent detection was performed using SuperSignal West Pico PLUS chemiluminescent substrate (34580, Thermo Fisher Scientific, Waltham, MA, USA) and C-DiGit blot scanner (LI-COR Biosciences, Linkoln, NE, USA) in a high-sensitivity mode (12-min scanning).

### 2.10. Statistical Analysis

Statistical analysis was performed using GraphPad Prism 8 (GraphPad Software, San Diego, CA, USA). For descriptive statistics, data are presented as proportions, median, 25th and 75th percentiles, and range. Two independent groups were compared by the Mann–Whitney U-test. Proportions were compared by Pearson’s chi-squared test with Yates’s correction for continuity. The *p* values ≤ 0.05 were regarded as statistically significant.

## 3. Results

### 3.1. Biodegradable Electrospun TEVGs Can Be Reinforced with Anti-Aneurysmatic PCL Sheath and Modified with Iloprost and 1,5-bis-(4-tetradecyl-1,4-diazoniabicyclo [2.2.2] octan-1-yl) Pentane Tetrabromide

To design a haemocompatible electrospun TEVG, we immobilised an anti-thrombotic coating on their luminal surface using iloprost, a synthetic prostacyclin (prostaglandin I2) analogue inhibiting platelet activation, aggregation, and adhesion and causing vasodilation through binding and activation of IP (prostacyclin or prostaglandin I2 receptor), EP2 (prostaglandin E2 receptor 2) and EP4 (prostaglandin E2 receptor 4) receptors. As iloprost also ameliorates endothelial dysfunction and hinders leukocyte adhesion to endothelial cells, it has been used to treat COVID-19-associated vasculopathy [51,52]. Several reports also suggested a mild fibrinolytic activity of iloprost [53,54,55,56]. Antimicrobial effect was furnished by a cationic amphiphilic drug (CAD) 1,5-bis-(4-tetradecyl-1,4-diazoniabicyclo [2.2.2] octan-1-yl) pentane tetrabromide, which has RNase activity in contrast to other CADs and therefore has higher antibacterial and antiviral properties [33,57,58,59]. Both iloprost and CAD were conjugated to the TEVG luminal surface using a poly(vinylpyrrolidone) (PVP) hydrogel. Electron microscopy analysis of such biofunctionalised TEVGs (PCL^Ilo/CAD^) revealed highly porous graft wall (Figure 1a) consisting of intertwined PCL fibers from 0.6 to 2.5 µm diameter which formed interconnected pores (Figure 1b). The outer surface of the TEVG was enforced with anti-aneurysmatic sheath by fused deposition modeling, an additive manufacturing process employing a continuous filament of a thermoplastic material (e.g., PCL), using an original 3D printer extruder (Figure 1c). Such sheath intimately attached to the TEVG and consisted of continuous and crossed PCL fibers up to 241 µm diameter (Figure 1c,d).

To characterise the PVP layer immobilised on the TEVG luminal surface, we applied Fourier-transform infrared spectroscopy (FTIR). The FTIR spectra of unmodified and PVP-modified TEVGs showed C=O stretching bands at 1724 cm^−1^ and C–O stretching bands at 1278 and 1054 cm^−1^ (Figure 2). In all samples, low-intensity bands of asymmetric vibrations in the CH_2_– group and symmetric vibrations in the CH_3_– group were observed at 2942 cm^−1^ and 2865 cm^−1^, respectively (Figure 2). In contrast to the unmodified TEVGs, the spectrum of the PVP-modified TEVGs contained a band at 1654 cm^–1^ corresponding to the –CONH_2_ group of the pyrrolidone ring, confirming that PVP was indeed conjugated with the TEVG surface (Figure 2). As the position and intensity of the FTIR bands upon conjugation with PVP did not change, we suggested a negligible effect of such modification on the PCL structure.

Next, we determined the amount of iloprost on the surface of the PCL^Ilo/CAD^ TEVGs and in PBS solution at ascending time points of incubation at 37 °C. A gradual increase in the iloprost level on TEVG surface was observed during the first 20 days of incubation, followed by a decrement and end of iloprost release after 3 months (Figure 3).

Cationic amphiphilic drugs (CADs) have a characteristic bromine peak at 1.5 keV (Figure 4a). Energy-dispersive X-ray spectroscopy of PCL^Ilo/CAD^ TEVGs successfully found bromine in all modified samples, confirming the attachment of 1,5-bis-(4-tetradecyl-1,4-diazoniabicyclo [2.2.2] octan-1-yl) pentane tetrabromide to the luminal surface (Figure 4b and Table 1). Storage at 4 °C for 1 day or at −20 °C for 1 day or 6 months did not affect the bromine peak (Figure 4c,d, and Table 1). However, 6 h incubation of TEVGs in PBS supplied with 0.05% Tween-20 at 37 °C removed the bromine peak from the luminal surface (Figure 4e and Table 1).

LC-MS analysis of PBS incubated with TEVGs did not reveal the presence of the soluble CADs. Mass spectrometry without the chromatographic separation also did not document the peaks associated with C_41_H_84_Br_3_N_4_^+^ molecular ion (m/z 871.42256) or amphiphile molecular ion with phosphate as a counterion in PBS samples, testifying to the stable conjugation of 1,5-bis-(4-tetradecyl-1,4-diazoniabicyclo [2.2.2] octan-1-yl) pentane tetrabromide with the TEVG luminal surface (Figure 5). However, positive control incubation in methanol at 40 °C for 6 h after 3-months’ incubation with PBS induced the peaks corresponding to phosphate salts of C_41_H_84_Br_3_N_4_^+^ molecular ion, which formed as a result of ion exchange with PBS, though a mentioned peak at m/z 871.42256 has not been detected (Appendix A).

### 3.2. PCL^Ilo/CAD^ TEVGs Are Endothelialised and Show Hierarchical Structure 6 Months upon the Implantation into the Ovine Carotid Artery

TEVGs have been implanted into the ovine carotid artery (n = 12) for 6 months, and all animals survived until the end of the follow-up. Doppler ultrasonography showed that primary patency was 83.3% (10/12 animals) 1 month postimplantation and 50.0% (6/12 animals) 3 and 6 months postimplantation, respectively. Six months postimplantation, gross examination demonstrated that the TEVGs fully retained their integrity and none of them suffered from aneurysms (Figure 6a). The luminal surface of the TEVGs was free from any clots, indicating excellent haemocompatibility provided by iloprost conjugation (Figure 6a). The PCL scaffold was completely resorbed and replaced with vascular tissue (Figure 6a), similar to the contralateral carotid artery (Figure 6b). The luminal surface of the TEVGs (Figure 6a) was visually similar to the contralateral healthy carotid artery (Figure 6b) and to the anastomotic sites containing non-absorbable suture material (Figure 6c).

We then conducted a microscopic examination of ovine carotid arteries and TEVGs. Ovine carotid arteries consisted of: (1) dense endothelial cell (EC) monolayer attached to the collagen IV-enriched basement membrane; (2) thick medial layer with multiple lamellar units which included elastic laminae and vascular smooth muscle cells (VSMCs) embedded into the collagen III/IV extracellular matrix between them; (3) thick collagen-rich tunica adventitia (Figure 7). Similar to human ECs, ovine carotid artery ECs were CD31- and von Willebrand factor (vWF)-positive and underlying VSMCs abundantly expressed α-smooth muscle actin (α-SMA, Figure 7). Tunica adventitia of the ovine carotid artery contained collagen bundles, numerous *vasa vasorum*, perivascular adipose cells, and fibroblasts (Figure 7). Hence, ovine and human carotid artery anatomies were considered as similar.

Six months postimplantation, TEVGs were completely resorbed and replaced with vascular tissue formed de novo (Figure 8). Similar to the contralateral ovine carotid artery, TEVGs (i.e., regenerated arteries are still termed TEVGs for simplicity) were composed of: (1) CD31- and vWF-positive EC monolayer without a clear basement membrane; (2) thick medial layer containing residual PCL and mesenchymal cells surrounded by abundant collagen fibers; (3) tunica adventitia with plenty of collagen bundles and vasa vasorum (Figure 8). However, in contrast to the native carotid arteries, TEVGs lacked elastic fibers and their tunica adventitia was heavily infiltrated by canonical macrophages and multinucleated giant cells in addition to fibroblasts (Figure 8). A thick blood vessel wall and absence of calcium deposits testified to the retained integrity of the TEVG and suggested synchronisation of polymer degradation and vascular tissue regeneration.

### 3.3. Proteomic Profiling of PCL^Ilo/CAD^ TEVGs and Contralateral Carotid Arteries Reveal Molecular Signatures of TEVG Remodeling

To compare the molecular composition of the native carotid arteries and TEVGs 6 months postimplantation, we performed a label-free, pairwise proteomic profiling (i.e., each TEVG and contralateral carotid artery have been collected from the same animal, 12 pairs in total) by means of ultra-high performance liquid chromatography–tandem mass spectrometry (UHPLC-MS/MS). A proteomic approach was preferred over the RNA sequencing as TEVGs might contain physiologically significant proteins emerging from the circulation rather than produced in situ.

Principal component analysis indicated a clear discrimination pattern between the native carotid arteries and TEVGs (71% explained variation), whereas the variation between the samples comprising each group was relatively low (8% explained variation, Figure 9a). The number of unique or significantly upregulated proteins was 145 in the native carotid arteries and 128 in the TEVGs (Figure 9b). When assessing the distribution of such proteins across the cellular compartments as annotated by COMPARTMENTS database [45], we revealed the significant overrepresentation of cytoskeleton proteins in the native carotid arteries (56 out of 145 (38.6%) unique or differentially expressed proteins) as compared to the TEVGs (26 of 128 (20.3%) unique or differentially expressed proteins). In contrast, lysosomal and endosomal proteins were abundant in the TEVGs (28/128 (21.9%) and 20/128 (15.6%) proteins, respectively, Figure 9c) in comparison with the native carotid arteries (8/145 (5.5%) and 5/145 proteins (3.4%), respectively, Figure 9c). Distribution of proteins related to other organelles did not show statistically significant differences (Figure 9c).

Bioinformatics analysis of the differential distribution of proteins over the cellular compartments using Gene Ontology (GO) Cellular Component (Table 2) and UniProtKB Keywords (Table 3) databases confirmed the abundance of the terms related to the cytoskeleton (i.e., stress and contractile fibers, focal adhesion, actin and total cytoskeleton) and basement membrane in the native carotid arteries, whilst TEVGs exhibited increased expression of lysosomal, endosomal, endoplasmic reticulum (in particular smooth endoplasmic reticulum), and endoplasmic reticulum–Golgi intermediate compartment proteins.

The following analysis of GO Molecular Functions (Table 4) and UniProtKB Keywords (Table 5) databases documented that proteins responsible for the laminin, integrin, actin, and cytoskeletal protein binding were enriched in native carotid arteries whereas those related to the proton-transporting ATPase activity, unfolded protein binding, chaperone binding, and protein folding prevailed in TEVGs, in keeping with the previous findings.

Further, bioinformatics analysis of differentially regulated biological processes employing GO Biological Process database revealed that actin cytoskeleton and basement membrane organization, actin filament depolymerization, stress fiber, myofibril, and focal adhesion assembly, integrin-mediated signaling pathway, muscle cell development, and muscle contraction were upregulated in the native carotid arteries (Table 6). TEVGs showed the signatures of glucose, pentose, nucleotide, NADP/NADPH, and collagen metabolism, energy generation, protein folding, and proteolysis regulation including acidification (Table 6).

Analysis of the Reactome database revealed concordant results, as the terms related to the integrin, syndecan, and laminin interactions, dermatan and chondroitin sulfate biosynthesis, fibronectin matrix formation, extracellular matrix proteoglycans, smooth muscle contraction, and elastic fiber formation were overrepresented in the native carotid arteries (Table 7). TEVGs demonstrated overexpression of the proteins participating in tricarboxylic acid cycle, pentose phosphate pathway, biosynthesis, assembly, formation and degradation of collagen, translation elongation, transport from the endoplasmic reticulum to the Golgi apparatus, and production of reactive oxygen/nitrogen species (Table 7).

A search of the Kyoto Encyclopedia of Genes and Genomes (KEGG) database also identified upregulated actin cytoskeleton regulation, focal adhesion, and VSMC contraction in the native carotid arteries as well as augmented lyso- and phagosomal activity and carbon/amino acid metabolism in TEVGs (Table 8).

Detailed analysis of unique and differentially expressed proteins identified an overrepresentation of the contractile proteins serving as VSMC markers (7/145), extracellular matrix proteins composing the basement membrane (16/145), and antioxidant defense proteins (5/145, including Cu-Zn and mitochondrial superoxide dismutase and glutathione S-transferase Mu) in the native carotid arteries (Figure 10). Strikingly, TEVGs were almost devoid of the most specific VSMC markers (i.e., smooth muscle myosin heavy chain and smoothelin), and the most abundant VSMC markers (i.e., calponin and transgelin/SM22α) showed a negligible relative expression in the TEVGs (Figure 10). The least fold change for the VSMC markers in the native carotid arteries as compared to TEVGs was around 10 (for α-SMA, Figure 10). In keeping with the immunostaining findings, key basement membrane components were also absent (such as laminin subunits and nidogen) or significantly downregulated (such as chondroitin sulfate and fibronectin) in TEVGs (Figure 10).

Expression of lysosomal proteins (19/128) such as cathepsins (B, D, H, K, S, and Z), V-type proton ATPase subunits (A, B, C1, E1, G1, and H), and lysosome-associated membrane glycoproteins (LAMP1 and LAMP2) were restricted to or vastly increased in TEVGs indicative of chronic inflammation, as also evidenced by exclusive expression of a specific macrophage marker CD68 (Figure 10). Upregulation of the proteins responsible for the protein synthesis and folding (14/128) and those accountable for glucose metabolism (15/128) testified to the active regeneration occurring during the replacement of PCL with newly formed vascular tissue (Figure 10).

To verify the proteomic profiling results, we performed ultrastructural visualisation of remodeled TEVGs and clearly distinguished macrophages and multinucleated giant cells in the tunica adventitia (Figure 11a). Western blotting measurements showed that, in keeping with UHPLC-MS/MS findings, specific macrophage marker CD68 was abundant in all measured TEVG samples (8/8) while intact contralateral carotid arteries were devoid of CD68 (Figure 11b). A similar trend was observed when measuring the key glycolytic enzyme glyceraldehyde 3-phosphate dehydrogenase (GAPDH, Figure 11b), which is frequently used as a housekeeping protein in Western blotting measurements, and therefore we also performed total protein normalisation by Coomassie Brilliant Blue staining (Figure 11b). Total counts of macrophages (CD68^+^ cells) and proportions of anti-inflammatory and pro-fibrotic (M2) macrophages, notable for CD163 expression, did not differ significantly between the non-occluded and occluded TEVGs (Figure 11c), suggesting that macrophages are not among the major contributors to the TEVG occlusion, in contrast to the insufficient compliance. Taken together, these findings confirmed macrophage-driven remodeling as a driving force of polymer biodegradation and vascular regeneration upon the TEVG implantation, also showing that TEVG occlusion does not rely on the macrophage-dependent fibrosis.

Collectively, electron microscopy, immunostaining, and proteomic profiling demonstrated that the processes of polymer degradation and vascular tissue regeneration in PCL^Ilo/CAD^ TEVGs were synchronised, as evident by the active energy metabolism and protein synthesis, folding, and assembly. However, such TEVGs lacked both elastic fibers and an organised VSMC layer, that points out the risks of aneurysms in the long term. Our TEVGs were also depleted of basement membrane components and did not display any basement membrane at the confocal microscopy upon anti-collagen IV staining, yet were fully endothelialised and resistant to thrombosis. In addition, PCL^Ilo/CAD^ TEVGs were notable for the considerable macrophage-driven inflammation (as evident by numerous lysosomal markers and in particular CD68 expression) and weaker antioxidant defense suggestive of a significant oxidative stress. Nevertheless, PCL^Ilo/CAD^ TEVGs did not show any signs or molecular signatures of thrombosis, calcification, atherosclerosis, or infection.

## 4. Discussion

Albeit, the development of commercially available off-the-shelf TEVG is highly relevant to the current demands of cardiac surgery, neurosurgery, and microsurgery and would significantly address a growing need in the increase in vascular reconstruction procedures, none of the TEVG prototypes has passed through the pre-clinical trials hitherto. Among the most frequent complications of TEVG implantation are thrombosis, aneurysms, neointimal hyperplasia, calcification, atherosclerosis, and infection [1,2,3,4,5,6,7,8]. Previously, our group reported an efficient approach for increasing haemocompatibility and thrombosis prevention by immobilising an anticoagulant heparin and a vasodilator/antiplatelet drug, iloprost, to the luminal surface of PCL-based TEVGs [25]. Further, we have developed an anti-aneurysmatic PCL sheath attached to the TEVG outer surface by fused deposition modeling, a method employing 3D printing extrusion [60]. Here, we combined iloprost and a cationic amphiphilic drug (CAD) 1,5-bis-(4-tetradecyl-1,4-diazoniabicyclo [2.2.2] octan-1-yl) pentane tetrabromide, which has higher antibacterial and antiviral properties in comparison with other CADs because of RNAse activity. All sheep survived after the TEVG implantation into the carotid artery and our TEVGs demonstrated a 50.0% primary patency (6/12 animals) 6 months postimplantation.

Despite as many as 182 animal studies assessing the efficiency of TEVG implantation and 22 pre-clinical trials of various TEVGs (12 in the ovine and 10 in the porcine model) being carried out to-date [61], a vast majority of them suffered from a limited molecular understanding of the regeneration processes following the implantation. To address this common and critical shortcoming significantly restricting further progress in TEVG design, we have, for the first time, performed a proteomic profiling of biodegradable TEVGs and intact contralateral carotid arteries. The idea behind this choice was that pairwise comparison of TEVGs and contralateral carotid arteries collected from the same animal is optimal to assess how close in situ vascular regeneration upon the TEVG implantation is to the corresponding intact artery within the same organism. By this design, we were able to evaluate the extent of tissue remodeling and the balance between physiological and pathological remodeling during the degradation of TEVGs and their replacement by regenerating vascular tissue, having the intact contralateral carotid artery as the gauge. The efficacy of iloprost has been shown in one of the previous papers by our group [25], and thence we focused on tissue remodeling and vascular regeneration features rather than on the modification efficiency.

Having considered protein categories overrepresented in native carotid arteries or TEVGs, we found VSMC markers, basement membrane components, and antioxidant defense proteins upregulated in the native carotid arteries and lysosomal markers, proteins responsible for the protein synthesis/folding, and glucose metabolism regulators overexpressed in TEVGs. Regarding the potential role of these categories in the long-term success of the TEVGs, VSMC contractile proteins mediate an adequate biomechanical response to the blood pressure changes through the controlled cyclic stretch, while basement membrane components contribute to the rapid and complete endothelialisation. Antioxidant defense proteins are probably in charge for the control of macrophage-driven inflammation, and their levels can be possibly corrected by an antioxidant-rich diet, such as the Mediterranean diet, in a clinical setting. Lysosomal enzymes and structural proteins (macrophage markers), components of the protein folding and synthesis machinery, and energy metabolism proteins are collectively responsible for the polymer degradation (requiring large amounts of cells enriched with acidic organelles and multiple enzymes to digest and recycle the polymer) and its replacement by the regenerative vascular tissue (demanding permanent protein supply to form the extracellular matrix), as both of these processes are extremely energy-consuming.

As opposed to the bioprosthetic heart valves (paper under revision, data deposited to the ProteomeXchange Consortium via the PRIDE partner repository with the dataset identifiers PXD035113 and 10.6019/PXD035113), TEVGs retained their integrity after a significant time postimplantation and we did not reveal the deposition of abundant serum proteins (e.g., albumin, angiotensinogen, fibrinogen, or lipoproteins) or matrix metalloproteinases in the vascular tissue replacing the degrading PCL. In combination with active biosynthetic processes, evident by increased amounts of the proteins participating in the translation, protein folding and assembly, and energy metabolism, these molecular patterns testified to the synchronisation between PCL degradation and vascular regeneration. Further, TEVGs did not demonstrate any signs of the infection such as precipitation of complement components or neutrophil markers, instead showing informative signatures of chronic inflammation remodeling, namely structural lysosomal proteins (which are well-established macrophage markers) and cathepsins, known as lysosomal but not blood- or extracellular matrix-derived enzymes [62,63,64]. This type of remodeling has already been reported in several studies [19,65,66,67,68], as well as the rapid degradation of the polymer scaffolds in sheep as compared to rats [25,34,69,70,71,72,73]. As synchronisation of polymer biodegradation and vascular regeneration evidently occurs 6 months postimplantation (the duration of follow-up in our study), and PCL is almost completely degraded and replaced with de novo vascular tissue, it seems unlikely that the TEVG will become considerably closer to the native carotid artery in the long term.

Therefore, the advantages of our PCL^Ilo/CAD^ TEVGs included controlled replacement of the biodegradable polymer by de novo vascular tissue and freedom from a number of main complications accompanying the TEVG implantation (thrombosis, calcification, atherosclerosis, and infection). Although PCL^Ilo/CAD^ TEVGs did not have a clear basement membrane, it did not hinder endothelialisation, a mandatory pre-requisite for haemocompatibility and thromboresistance [1,2,3,4,5,6,7,8], and a CD31- and vWF-positive EC monolayer has been observed. The most significant drawback was the lack of the hierarchical arterial medial structure which is recognisable by elastic fibers and contractile VSMCs that form lamellar units in the elastic arteries, such as the carotid artery. This could potentially lead to the insufficient compliance, i.e., an inadequate biomechanical response, such as contraction or distension, to the changes of the systolic and diastolic blood pressure, respectively. The absence of elastic fibers is a considerable and general disadvantage of biodegradable TEVGs, albeit a recent study reported the successful formation of the internal elastic lamina by the embedding of tropoelastin fibers into the poly(glycerol sebacate), a highly elastic and degradable biomaterial, during electrospinning [74]. Yet, the enclosure of poly(glycerol sebacate) or fibrin scaffold into the PCL [75,76] or poly(glycolic acid) [77] sheath efficiently prevented the development of aneurysms, similar to our previous work [60]. The lack of contractile VSMCs under significant cyclic strain characteristic of the ovine carotid artery was unexpected, as this is a major stimulus of such molecular specification [78,79,80]. Another shortcoming observed in TEVGs was the downregulation of antioxidant defense, yet it most likely accompanied the macrophage-driven inflammation rather than represented a separate phenomenon.

These results indicate two main issues remaining unresolved for the ultimate success in pre-clinical trials of bioresorbable TEVGs. The first, an inability to guide the assembly and formation of regular elastic fibers, is apparent but is probably not a major problem in the reconstruction of arteries with low shear stress (e.g., in the coronary artery bypass graft surgery and microsurgery settings). Even if the internal elastic lamina can be reproduced using certain composites, as was claimed in [74], it would barely supply the TEVGs with the biomechanical features of the elastic arteries consisting of multiple lamellar units. The absence of the basement membrane does not preclude endothelialisation of TEVGs, as was shown in this study, and therefore is likely not a major drawback, since the primary function of the basement membrane is to provide an attachment site for circulating endothelial cells, endothelial progenitor cells, and endothelial cells crawling from the anastomoses. However, our TEVGs have been completely endothelialised despite their length (4 cm, 10-fold higher than the 4 mm diameter) and absence of the clear basement membrane at fluorescence microscopy, as well as the dearth of the basement membrane components at proteomic profiling analysis.

The second problem—the lack of contractile VSMCs in the medial layer—seems to have a higher priority, as the recapitulation of VSMC differentiation in vivo remains challenging. Potential approaches to stimulate physiological, contractile VSMC specification of mesenchymal cells at cyclic stretch in the TEVG tunica media include dietary, vascular tissue engineering, and pharmacological interventions. First, a specific diet including recently defined factors of cardiovascular resilience such as calorie restriction, intermittent fasting, meal timing (i.e., adjusted diurnal rhythm of feeding), antioxidant-rich diet, and increased physical activity [81,82] might reduce the cardiovascular risk factor burden, retard the developing vascular occlusion, and improve the implantation outcomes. Intake of pre-, pro-, or synbiotics to manage a physiological gut microbiota profile might have synergistic effects with a healthy dietary pattern in the cardiovascular context [83,84,85,86,87]. Yet, the impact of pharmacological or nutritional interventions on the outcomes of TEVG implantation has not been evaluated to-date. Second, direction of mesenchymal cells towards the vascular smooth muscle fate requires an adequate biomechanical response to the blood flow since the first hours postimplantation, and optimisation of anti-aneurysmatic sheath thickness (sufficiently high to prevent aneurysms but still allowing physiological movement of the TEVG) is another way to increase early vascular compliance and elastance at the implantation site. Third, immobilisation of anti-microtubule agents such as paclitaxel or mTOR inhibitors (sirolimus, everolimus, or biolimus) might significantly inhibit neointimal hyperplasia and therefore provide a time frame for the VSMC layer formation [88].

We propose that there is a correlation between the flow parameters (such as flow pattern, shear stress, or distribution of haemodynamic load) and VSMC specification, and regular measurements of the mentioned characteristics in TEVGs by means of various visualisation modalities can pave the way to our understanding of the transition between synthetic mesenchymal cells and contractile, fully functioning VSMCs. In other words, the key for the TEVG success currently lies within the medial layer. Possibly, TEVGs should be first tested in the position of muscular (e.g., coronary or internal thoracic artery) but not elastic artery; the differences between the results of the TEVG implantation into artery or vein position have been convincingly described earlier [66].

As shown in this study, unbiased transcriptomic or proteomic approaches are capable of gaining an insight into the vascular regeneration upon TEVG implantation and should be extensively employed in the pre-clinical trials of TEVGs to improve their design and testing settings. Further directions to apply high-throughput molecular characterisation in relation to TEVGs might include a detailed profiling of the EC monolayer, as its proper functioning is crucially important for providing acceptable primary patency [89,90,91]. The progress in the development of TEVGs largely depends on the molecular understanding of their in situ remodeling and decoding of molecular signatures associated with physiological and pathological vascular regeneration during and upon the polymer replacement. This corresponds to the recent trends in the field [29], although vascular tissue engineers primarily focus their efforts on harnessing novel visualisation modalities (such as intravascular ultrasound, 3D fast spin-echo T1 black-blood imaging, contrast-enhanced 3D magnetic resonance angiography, time-resolved magnetic resonance 2D and 3D flow velocity mapping, and delayed enhancement magnetic resonance imaging) as well as on employing computational modeling of fluid–structure interactions in context of subject-specific 3D TEVG geometries. Yet, we believe that the following studies should extensively employ transcriptomic and proteomic approaches to characterise the gene and protein expression in TEVGs at ascending time points, identify the molecular signatures of various vascular regeneration patterns, pinpoint the molecular events behind the polymer degradation and replacement in the anastomotic sites and in the midgraft, and evaluate the distance between in situ regenerated blood vessels and contralateral arteries. Hopefully, the combination of pharmacological approaches (conjugation with heparin, iloprost, CADs, or paclitaxel/sirolimus/everolimus), appropriate site of the implantation (muscular artery of sheep or swine), high-resolution visualisation techniques, and computational modeling of the flow velocity and distribution will eventually result in the development of clinically applicable TEVGs which will be able to pass the pre-clinical trials successfully.

## Figures and Tables

**Figure 1 polymers-14-05149-f001:**
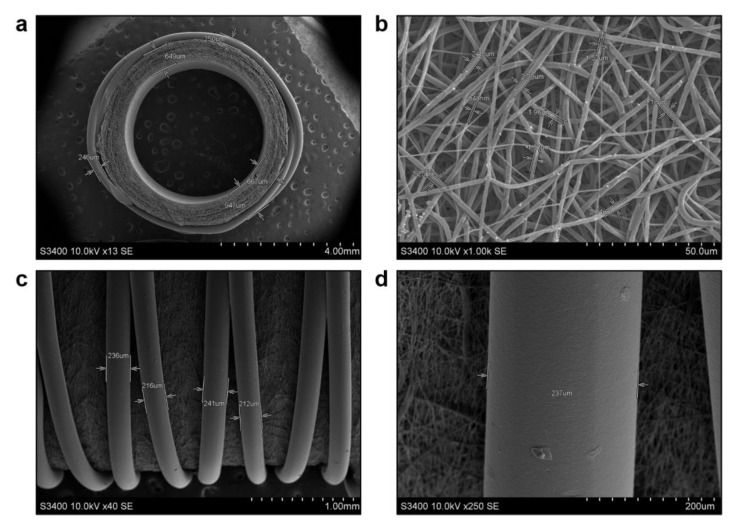
Electron microscopy examination of electrospun PCL^Ilo/CAD^ TEVGs before their implantation. (**a**) Tubular scaffold with a highly porous wall and anti-aneurysmatic PCL sheath, ×13 magnification, scale bar: 4 mm; (**b**) Graft wall consisting of micro- to nanoscale fibers and interconnected pores, ×1000 magnification, scale bar: 50 µm; (**c**) Large, continuous and partially crossed filaments of an anti-aneurysmatic PCL sheath, ×40 magnification, scale bar: 1 mm; (**d**) A close-up of the filament within an anti-aneurysmatic PCL sheath, ×250 magnification, scale bar: 200 µm. Accelerating voltage: 10 kV.

**Figure 2 polymers-14-05149-f002:**
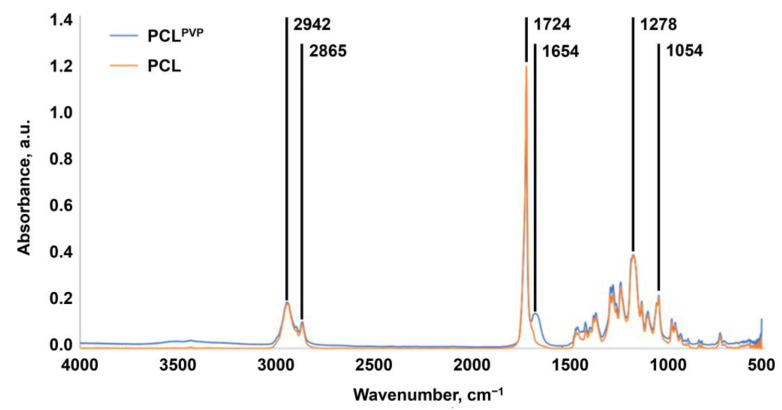
Fourier-transform infrared spectroscopy of unmodified and PVP-coated TEVGs. Note C=O stretching bands at 1724 cm^−1^, C–O stretching bands at 1278 and 1054 cm^−1^, asymmetric vibrations in the CH_2_– group at 2942 cm^−1^, symmetric vibrations in the CH_3_– group at 2865 cm^−1^, and –CONH_2_ group within the pyrrolidone ring at 1654 cm^−1^. The latter functional group is unique for PVP-coated TEVGs.

**Figure 3 polymers-14-05149-f003:**
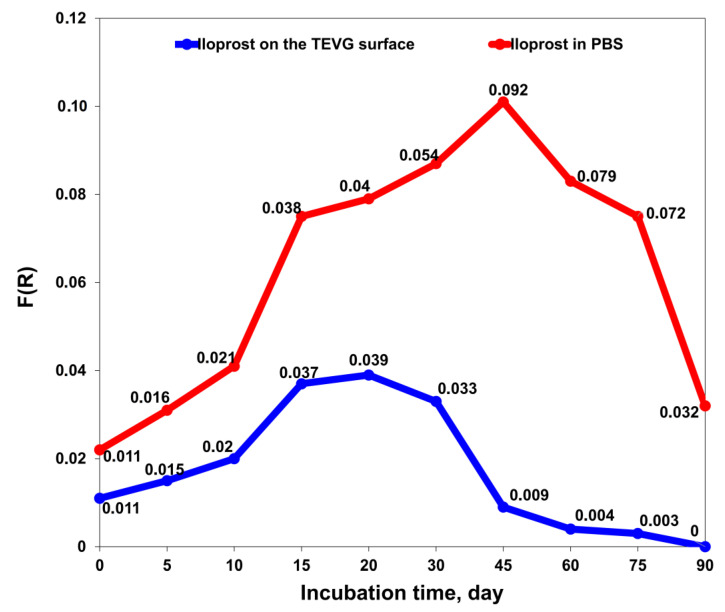
Release kinetics of iloprost from PCL^Ilo/CAD^ TEVGs over 90 days measured by Fourier-transform infrared spectroscopy. The graph represents iloprost content on the TEVG luminal surface (blue line and dots) and in phosphate-buffered saline (red line and dots) at ascending time points of incubation (1, 5, 10, 15, 20, 30, 45, 60, 75 and 90 days).

**Figure 4 polymers-14-05149-f004:**
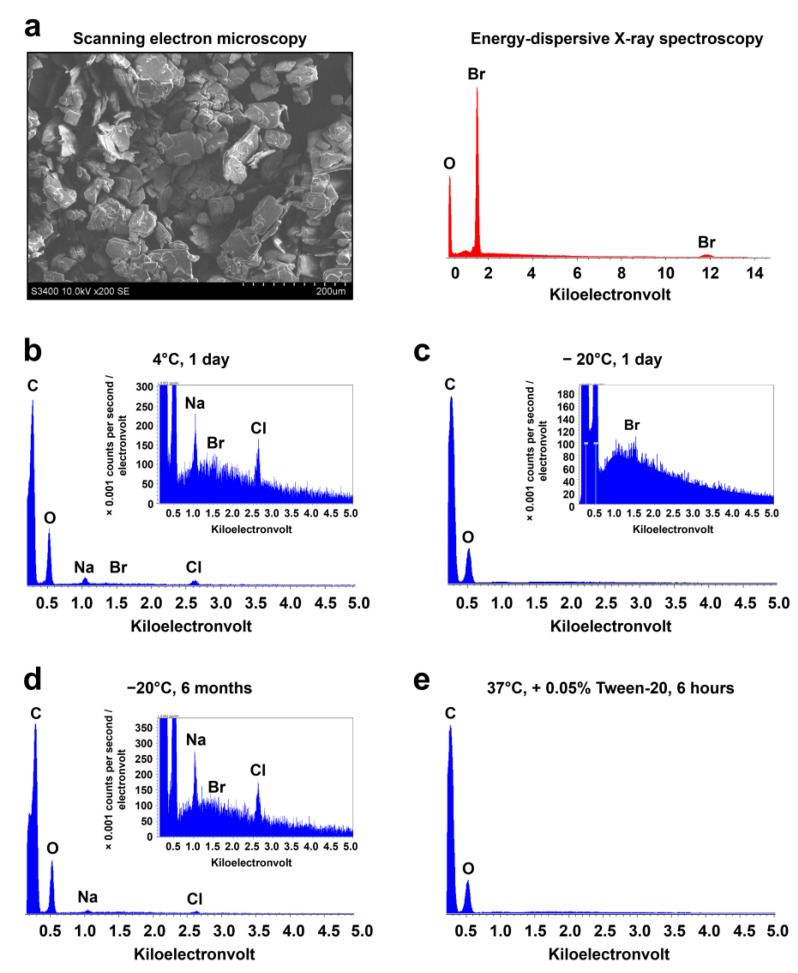
Chemical characterisation of the cationic amphiphilic drug 1,5-bis-(4-tetradecyl-1,4-diazoniabicyclo [2.2.2] octan-1-yl) pentane tetrabromide attached to the TEVG. (**a**) Scanning electron microscopy and energy-dispersive X-ray spectroscopy (EDX) of free 1,5-bis-(4-tetradecyl-1,4-diazoniabicyclo [2.2.2] octan-1-yl) pentane tetrabromide; (**b**) EDX spectra of PCL^Ilo/CAD^ TEVGs stored at 4 °C for 24 h; (**c**) EDX spectra of PCL^Ilo/CAD^ TEVGs stored at −20 °C for 1 day; (**d**) EDX spectra of PCL^Ilo/CAD^ TEVGs stored at −20 °C for 6 months; (**e**) EDX spectra of PCL^Ilo/CAD^ TEVGs incubated in PBS supplied with 0.05% Tween-20 at 37 °C for 6 h. Note the absence of the bromine peak in the latter experimental conditions.

**Figure 5 polymers-14-05149-f005:**
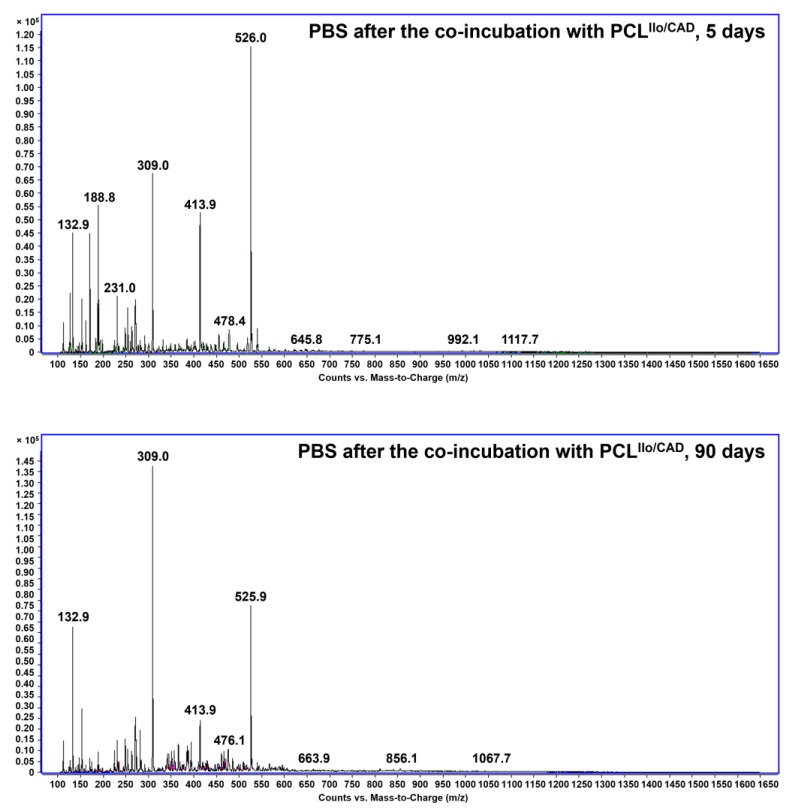
Liquid chromatography-mass spectrometry analysis of PBS samples after incubation with PCL^Ilo/CAD^ TEVGs for 5 (top) or 90 (bottom) days.

**Figure 6 polymers-14-05149-f006:**
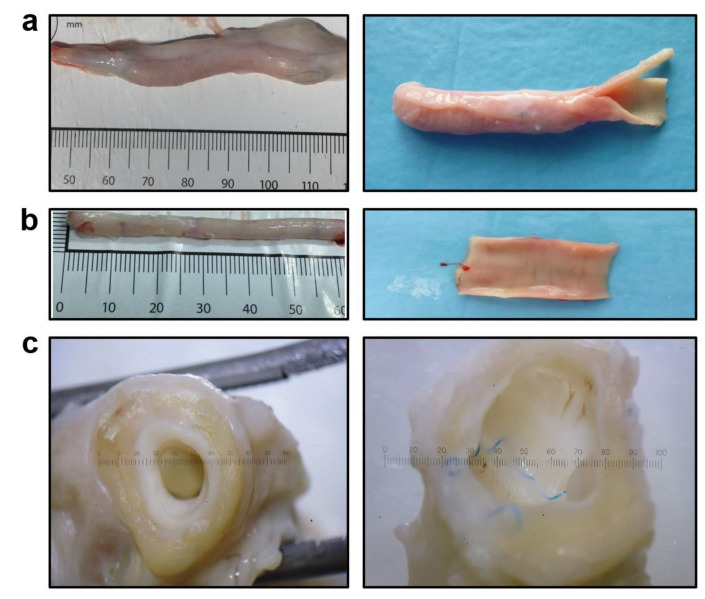
Gross examination of PCL^Ilo/CAD^ TEVGs and contralateral healthy carotid arteries 6 months postimplantation. (**a**) PCL^Ilo/CAD^ TEVG, intact (left) and longitudinal section to show the luminal surface (right), ×10 magnification; (**b**) Contralateral healthy carotid artery, intact (left) and longitudinal section to show the luminal surface (right), ×10 magnification; (**c**) A cross-section view showing the similarity of TEVG and anastomotic site luminal surfaces.

**Figure 7 polymers-14-05149-f007:**
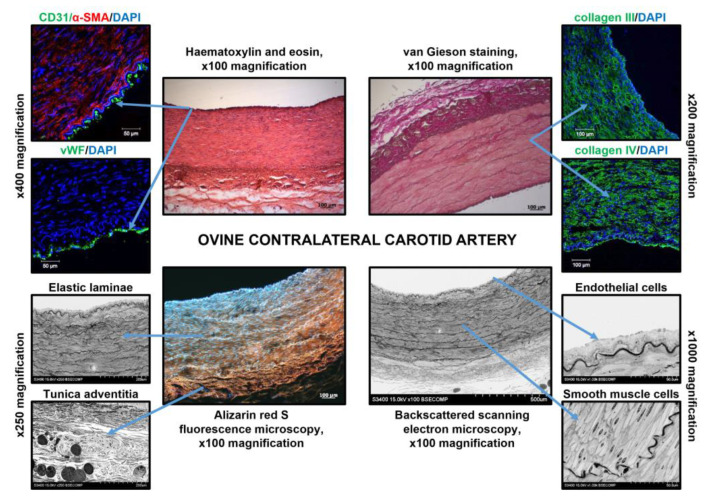
Microscopic examination of the ovine contralateral carotid artery. Centre: haematoxylin and eosin staining, van Gieson staining, alizarin red S staining, and EM-BSEM analysis, ×100 magnification, scale bar: 100 µm (for EM-BSEM analysis, scale bar: 500 µm). Left top: immunofluorescence staining for ECs (CD31 or vWF) and VSMCs (α-SMA), ×400 magnification, scale bar: 50 µm. Right top: immunofluorescence staining for medial collagens (type III and type IV), ×200 magnification, scale bar: 100 µm. Nuclei are counterstained with DAPI. Left bottom: EM-BSEM analysis, lamellar units and tunica adventitia, ×250 magnification, scale bar: 200 µm. Right bottom: EM-BSEM analysis, ECs and VSMCs, ×1000 magnification, scale bar: 50 µm. Note the EC monolayer, regular lamellar units (elastic laminae and VSMCs between them), and collagen-rich tunica adventitia.

**Figure 8 polymers-14-05149-f008:**
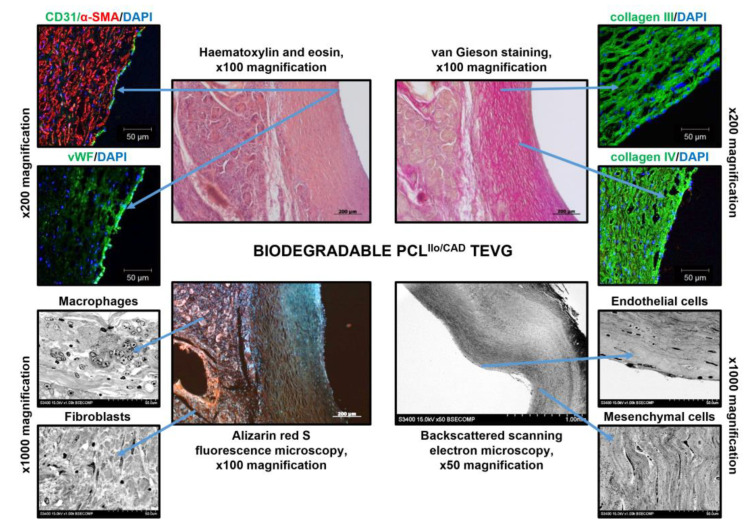
Microscopic examination of the TEVG implanted into the ovine carotid artery for 6 months. Centre: haematoxylin and eosin staining, van Gieson staining, and alizarin red S staining, ×100 magnification, scale bar: 200 µm (for EM-BSEM analysis: ×50 magnification, scale bar: 1000 µm). Left top: immunofluorescence staining for ECs (CD31 or vWF) and VSMCs (α-SMA), ×400 magnification, scale bar: 50 µm. Right top: immunofluorescence staining for medial collagens (type III and type IV), ×400 magnification, scale bar: 50 µm. Nuclei are counterstained with DAPI. Left bottom: EM-BSEM analysis, macrophages and fibroblasts, ×1000 magnification, scale bar: 50 µm. Right bottom: ECs and mesenchymal cells, ×1000 magnification, scale bar: 50 µm.

**Figure 9 polymers-14-05149-f009:**
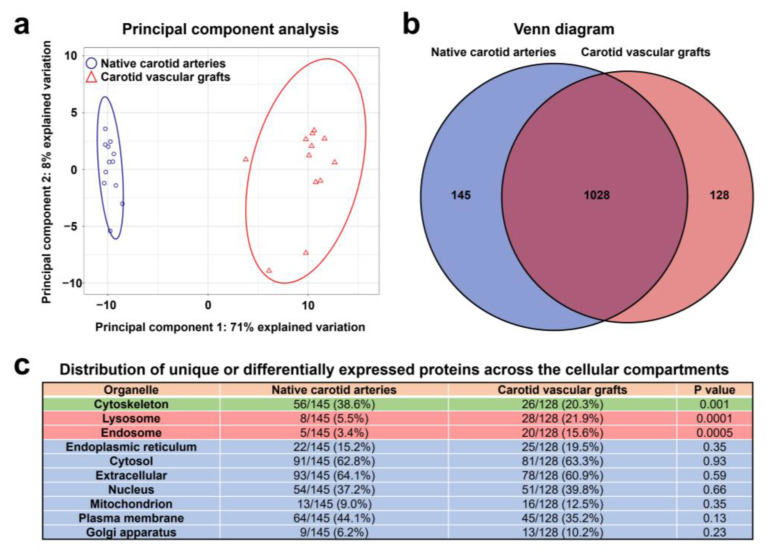
Shotgun proteomics analysis of TEVGs and contralateral carotid arteries by UHPLC-MS/MS. (**a**) Principal component analysis demonstrates clear discrimination of native carotid arteries (blue circles) and TEVGs (red triangles) implanted into the ovine carotid artery for 6 months, each point represents one sample. (**b**) Venn diagram showing the number of proteins unique or significantly overexpressed in the native carotid arteries (145) and TEVGs (128). (**c**) Distribution of differentially expressed proteins across the cellular compartments (annotated according to the COMPARTMENTS database); note the overrepresentation of cytoskeleton proteins in the native carotid arteries and abundance of lysosomal and endosomal proteins in the TEVGs.

**Figure 10 polymers-14-05149-f010:**
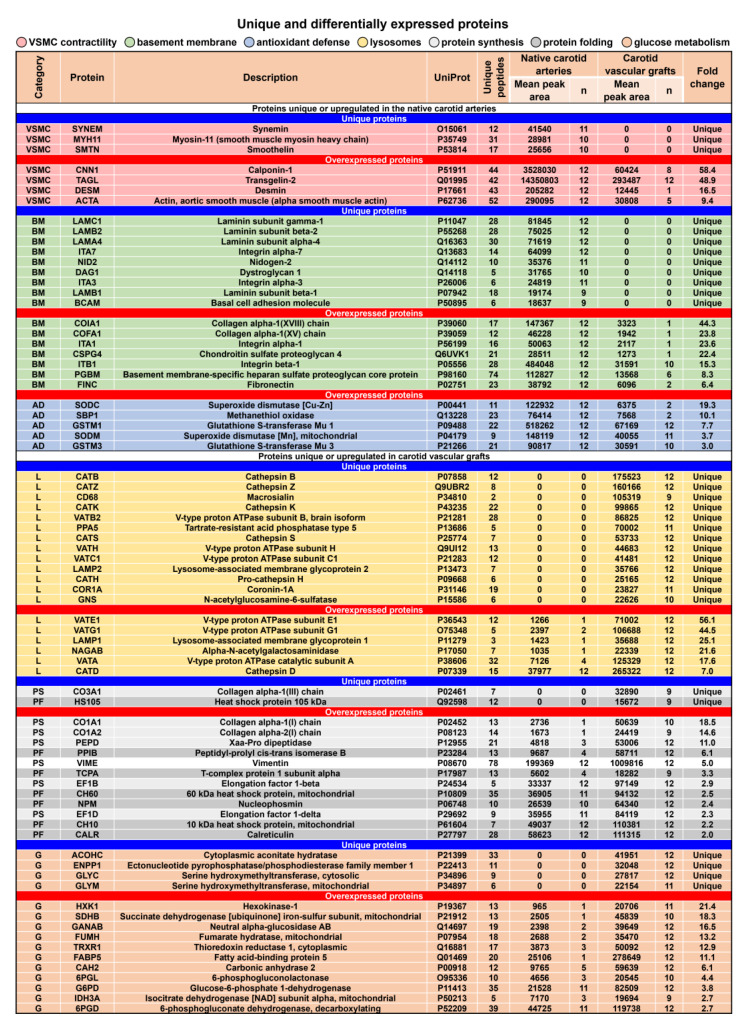
Unique and differentially expressed proteins in the native carotid arteries and TEVGs. Note the overrepresentation of contractile markers representing VSMC markers (pink colour), basement membrane components (green colour), and antioxidant defense proteins (blue colour) in the native carotid arteries and upregulation of lysosomal markers (yellow colour), proteins responsible for the protein synthesis (white colour) and folding (gray colour), and glucose metabolism regulators (orange colour) in TEVGs. For the proteins upregulated in the native carotid arteries, fold change is calculated as compared with TEVGs; for the proteins overexpressed in TEVGs, fold change is determined as compared with the native carotid arteries.

**Figure 11 polymers-14-05149-f011:**
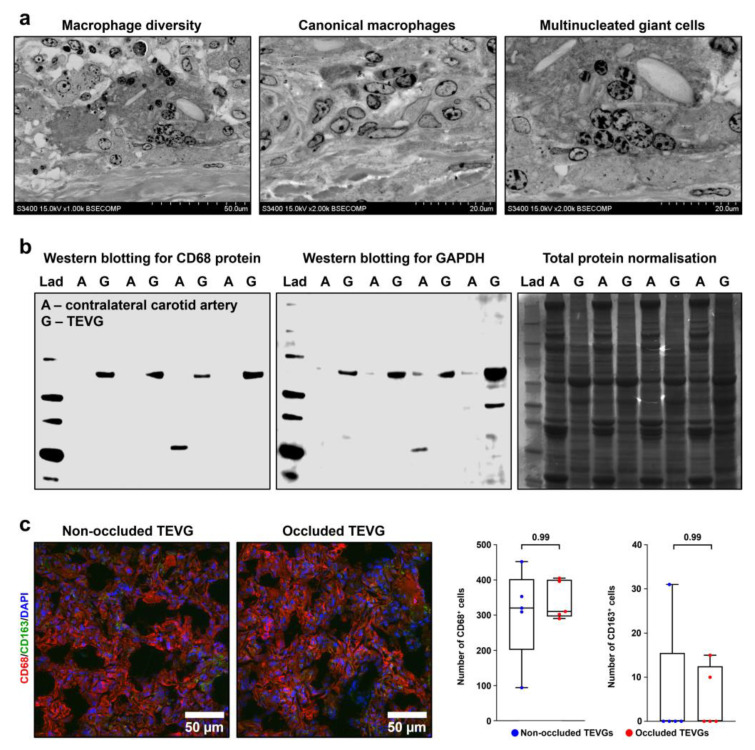
TEVGs but not intact contralateral carotid arteries are infiltrated by macrophages. (**a**) Backscattered scanning electron microscopy (EM-BSEM) shows massive macrophage invasion within the tunica adventitia of the TEVG. Left: overview of invading macrophages; centre: canonical macrophages; right: multinucleated giant cells. Magnification: ×1000 (left) and ×2000 (centre and right), scale bar: 50 and 20 µm respectively, accelerating voltage: 15 kV. (**b**) Western blotting of TEVG and contralateral carotid artery lysate for CD68 protein, representative image (left) and for glyceraldehyde 3-phosphate dehydrogenase (GAPDH), representative image (centre), and loading control by Coomassie Brilliant Blue staining (right). (**c**) Immunofluorescence staining for the pan-macrophage marker CD68 (red colour) and M2 macrophage marker CD163 (anti-inflammatory and pro-fibrotic macrophage specification, green colour). Magnification: ×400, scale bar: 50 µm. Nuclei are counterstained with DAPI. Representative images (left) and semi-quantitative analysis (right). Each dot on the plots represents the measurement from one image (n = 5 images per group). Whiskers indicate the range, box bounds indicate the 25th–75th percentiles, and centre lines indicate the median. *p* values are provided above boxes, Mann–Whitney U test.

**Table 1 polymers-14-05149-t001:** Elemental composition of PCL^Ilo/CAD^ TEVGs.

Incubation time	T, °C	Conditions	Br (wt.%)	O (wt.%)	Na (wt.%)	Cl (wt.%)
24 h	5	Without PBS	2.84	97.16	0.0	0.0
24 h	−20	Without PBS	2.75	97.25	0.0	0.0
6 months	−20	Without PBS	1.39	93.62	3.82	1.17
24 h	37	PBS + 0.05% Tween-20	0.0	100.0	0.0	0.0

**Table 2 polymers-14-05149-t002:** Distribution of proteins which are upregulated in either native carotid arteries or TEVGs across various cellular compartments (annotated according to the Gene Ontology Cellular Component database).

Gene Ontology Term	Total Proteins	Differentially Expressed Proteins	Fold Enrichment	FDR-Corrected *p* Value
Native carotid arteries				
Stress fiber	72	17	33.8	4.11 × 10^−18^
Contractile fiber	244	31	18.2	2.36 × 10^−26^
Basement membrane	101	11	15.6	2.14 × 10^−8^
Focal adhesion	423	41	13.9	2.65 × 10^−31^
Actin cytoskeleton	510	37	10.4	4.78 × 10^−24^
Collagen-containing extracellular matrix	434	24	7.9	8.82 × 10^−13^
Extracellular matrix	575	24	6.0	2.50 × 10^−10^
Extracellular vesicle	2123	82	5.5	1.96 × 10^−39^
Cytoskeleton	2401	57	3.4	1.40 × 10^−15^
Extracellular region	4396	92	3.0	1.95 × 10^−25^
Carotid vascular grafts				
Smooth endoplasmic reticulum	32	3	15.1	2.59 × 10^−2^
Endoplasmic reticulum-Golgi intermediate compartment	135	6	7.1	6.11 × 10^−3^
Collagen-containing extracellular matrix	434	15	5.6	5.95 × 10^−6^
Lysosome	749	28	6.0	4.49 × 10^−12^
Coated vesicle	310	11	5.7	1.88 × 10^−4^
Extracellular vesicle	2123	70	5.3	5.60 × 10^−32^
Extracellular matrix	575	15	4.2	1.53 × 10^−4^
Endosome	1036	22	3.4	2.35 × 10^−5^
Extracellular region	4396	79	2.9	3.00 × 10^−20^
Endoplasmic reticulum	2045	29	2.3	6.96 × 10^−4^

**Table 3 polymers-14-05149-t003:** Distribution of proteins which are upregulated in either native carotid arteries or TEVGs across various cellular compartments (annotated according to the UniProtKB Keywords database).

UniProtKB Keywords Term	Total Proteins	Differentially Expressed Proteins	Fold Enrichment	FDR-Corrected *p* Value
Native carotid arteries				
Basement membrane	7	4.6	22.0	3.2 × 10^−6^
Extracellular matrix	14	9.3	6.9	6.7 × 10^−7^
Cytoskeleton	41	27.2	4.2	4.0 × 10^−14^
Cell junction	24	15.9	3.7	6.1 × 10^−7^
Cytoplasm	88	58.3	2.2	4.1 × 10^−16^
Carotid vascular grafts				
Lysosome	16	12.5	6.3	8.8 × 10^−7^
Cytoplasmic vesicle	15	11.7	3.2	2.2 × 10^−3^
Cell junction	14	10.9	2.4	3.4 × 10^−2^
Cytoskeleton	20	15.6	2.2	9.6 × 10^−3^
Cytoplasm	66	51.6	1.8	8.8 × 10^−7^

**Table 4 polymers-14-05149-t004:** Molecular functions of proteins which are upregulated in either native carotid arteries or TEVGs (annotated according to the Gene Ontology Molecular Function database).

Gene Ontology Term	Total Proteins	Differentially Expressed Proteins	Fold Enrichment	FDR-Corrected *p* Value
Native carotid arteries				
Laminin binding	28	5	25.5	8.45 × 10^−4^
Integrin binding	159	15	13.5	7.24 × 10^−10^
Actin binding	451	35	11.1	4.67 × 10^−22^
Collagen binding	70	5	10.2	2.14 × 10^−2^
Cytoskeletal protein binding	1006	44	6.2	1.78 × 10^−19^
Carotid vascular grafts				
Proton-transporting ATPase activity, rotational mechanism	24	6	40.2	1.85 × 10^−5^
Collagen binding	70	6	13.8	1.57 × 10^−3^
Protein folding chaperone	68	5	11.8	1.31 × 10^−2^
Unfolded protein binding	124	7	9.1	3.30 × 10^−3^
Chaperone binding	107	6	9.0	1.21 × 10^−2^

**Table 5 polymers-14-05149-t005:** Molecular functions of proteins which are upregulated in either native carotid arteries or TEVGs (annotated according to the UniProtKB Keywords database).

UniProtKB Keywords Term	Total Proteins	Differentially Expressed Proteins	Fold Enrichment	FDR-Corrected *p* Value
Native carotid arteries				
Muscle protein	13	8.6	24.2	2.8 × 10^−12^
Integrin	5	3.3	12.9	6.2 × 10^−3^
Actin-binding	24	15.9	9.9	3.7 × 10^−15^
Oxidoreductase	15	9.9	2.9	6.2 × 10^−3^
Carotid vascular grafts				
Actin capping	4	3.1	26.4	7.0 × 10^−3^
Chaperone	10	7.8	5.6	2.1 × 10^−3^
Actin-binding	11	8.6	4.8	2.1 × 10^−3^

**Table 6 polymers-14-05149-t006:** Biological processes involving the proteins which are upregulated in either native carotid arteries or TEVGs (annotated according to the Gene Ontology Biological Process database).

Gene Ontology Term	Total Proteins	Differentially Expressed Proteins	Fold Enrichment	FDR-Corrected *p* Value
Native carotid arteries				
Actin filament depolymerization	12	4	47.7	7.46 × 10^−4^
Basement membrane organization	30	5	23.8	8.01 × 10^−4^
Stress fiber assembly	18	3	23.8	3.23 × 10^−2^
Myofibril assembly	62	10	23.1	5.04 × 10^−8^
Focal adhesion assembly	28	4	20.4	7.90 × 10^−3^
Extracellular matrix assembly	35	4	16.3	1.54 × 10^−2^
Endothelial cell development	51	5	14.0	5.61 × 10^−3^
Integrin-mediated signaling pathway	101	9	12.7	2.35 × 10^−5^
Muscle cell development	160	14	12.5	1.79 × 10^−8^
Cellular detoxification	108	8	10.6	3.52 × 10^−4^
Muscle contraction	230	15	9.3	1.15 × 10^−7^
Actin cytoskeleton organization	549	30	7.8	2.53 × 10^−14^
Cell-cell adhesion	543	20	5.3	1.14 × 10^−6^
Extracellular matrix organization	280	10	5.1	4.99 × 10^−3^
Cytoskeleton organization	1240	39	4.5	5.38 × 10^−12^
Carotid vascular grafts				
NADPH regeneration	16	4	40.2	2.33 × 10^−3^
Pentose metabolic process	12	3	40.2	1.67 × 10^−2^
Glucose 6-phosphate metabolic process	23	5	35.0	4.58 × 10^−4^
Tricarboxylic acid cycle	31	4	20.8	1.26 × 10^−2^
Collagen metabolic process	61	7	18.5	2.94 × 10^−4^
NADP metabolic process	37	4	17.4	1.89 × 10^−2^
Collagen catabolic process	41	4	15.7	2.32 × 10^−2^
Intracellular pH reduction	50	4	12.9	3.88 × 10^−2^
Chaperone-mediated protein folding	71	5	11.3	1.79 × 10^−2^
Response to unfolded protein	123	6	7.8	2.16 × 10^−2^
Protein folding	221	9	6.5	4.17 × 10^−3^
Generation of precursor metabolites and energy	406	12	4.7	3.74 × 10^−3^
Carbohydrate metabolic process	475	11	3.7	2.76 × 10^−2^
Carbohydrate derivative metabolic process	1005	22	3.5	3.50 × 10^−4^
Regulation of proteolysis	743	14	3.0	3.04 × 10^−2^

**Table 7 polymers-14-05149-t007:** Biological processes involving the proteins which are upregulated in either native carotid arteries or TEVGs (annotated according to the Reactome database).

Reactome Term	Total Proteins	Differentially Expressed Proteins	Fold Enrichment	FDR-Corrected *p* Value
Native carotid arteries				
Fibronectin matrix formation	6	3	71.5	1.80 × 10^−3^
Smooth muscle contraction	39	16	58.7	7.34 × 10^−19^
Cell–extracellular matrix interactions	18	7	55.6	1.04 × 10^−7^
Laminin interactions	30	11	52.4	2.70 × 10^−12^
Protein repair	6	2	47.7	4.07 × 10^−2^
Dermatan sulfate biosynthesis	11	3	39.0	5.86 × 10^−3^
Chondroitin sulfate biosynthesis	19	3	22.6	1.79 × 10^−2^
Non-integrin membrane-ECM interactions	59	9	21.8	2.82 × 10^−7^
ECM proteoglycans	76	11	20.7	1.10 × 10^−8^
Integrin signaling	26	3	16.5	3.49 × 10^−2^
Syndecan interactions	27	3	15.9	3.76 × 10^−2^
Elastic fibre formation	45	5	15.9	1.72 × 10^−3^
Molecules associated with elastic fibres	38	4	15.0	9.95 × 10^−3^
Integrin cell surface interactions	84	8	13.6	3.95 × 10^−5^
Cell junction organization	91	8	12.6	5.87 × 10^−5^
Carotid vascular grafts				
Pentose phosphate pathway	15	4	42.9	6.58 × 10^−4^
Insulin receptor recycling	26	6	37.1	1.63 × 10^−5^
ROS and RNS production in phagocytes	36	6	26.8	4.90 × 10^−5^
Citric acid cycle (TCA cycle)	22	3	21.9	2.49 × 10^−2^
Assembly of collagen fibrils and other multimeric structures	60	6	16.1	5.17 × 10^−4^
Collagen degradation	64	6	15.1	6.50 × 10^−4^
Collagen formation	89	8	14.5	3.87 × 10^−5^
Collagen biosynthesis and modifying enzymes	67	5	12.0	6.95 × 10^−3^
Trans-Golgi network vesicle budding	72	5	11.2	8.98 × 10^−3^
Degradation of the extracellular matrix	140	8	9.2	5.71 × 10^−4^
Eukaryotic translation elongation	94	5	8.6	2.03 × 10^−2^
Endoplasmic reticulum to Golgi anterograde transport	155	6	6.2	2.54 × 10^−2^
Asparagine N-linked glycosylation	305	10	5.3	2.56 × 10^−3^
Transport to the Golgi and subsequent modification	186	6	5.2	4.94 × 10^−2^
Membrane trafficking	626	12	3.1	2.95 × 10^−2^

**Table 8 polymers-14-05149-t008:** Biological processes involving the proteins which are upregulated in either native carotid arteries or TEVGs (annotated according to the Kyoto Encyclopedia of Genes and Genomes database).

Kyoto Encyclopedia of Genes and Genomes Term	Total Proteins	Differentially Expressed Proteins	Fold Enrichment	FDR-Corrected *p* Value
Native carotid arteries				
ECM–receptor interaction	12	7.9	13.4	5.8 × 10^−8^
Focal adhesion	24	15.9	11.7	1.5 × 10^−16^
Vascular smooth muscle contraction	10	6.6	7.3	1.3 × 10^−4^
Regulation of actin cytoskeleton	16	10.6	7.2	9.6 × 10^−8^
Carotid vascular grafts				
Lysosome	14	10.9	9.3	4.3 × 10^−7^
Carbon metabolism	11	8.6	8.4	3.8 × 10^−5^
Phagosome	13	10.2	7.5	1.1 × 10^−5^
Biosynthesis of amino acids	6	4.7	7.0	3.0 × 10^−2^
Metabolic pathways	38	29.7	2.2	7.2 × 10^−5^

## Data Availability

The mass spectrometry proteomics data have been deposited to the ProteomeXchange Consortium via the PRIDE partner repository with the dataset identifier PXD036520. Other data presented in this study are available on request from the corresponding author.

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
