# Peer review of "Controlled and Synchronised Vascular Regeneration upon the Implantation of Iloprost- and Cationic Amphiphilic Drugs-Conjugated Tissue-Engineered Vascular Grafts into the Ovine Carotid Artery: A Proteomics-Empowered Study"

_polymers, 2022, doi:10.3390/polym14235149_

Round 1

Reviewer 1 Report

Dear Authors

Thanks for you valuable work. I have only three comments that i think can improve your paper's quality.

1-     The material and method section should come after introduction.

2-     Please clarify the 3d printing parameters that has been used.

3-     Please present your tables in a more attractive way

Author Response

We sincerely thank the reviewer for the constructive criticism and valuable notes, which collectively helped us to improve the paper. Please see the attachment.

Reviewer 2 Report

This is an interesting work showing proteomic analysis of conjugated TEVGs in ovine models. With a few amendments/experimentations I recommend this for publication.

Major

- My main concern is the control group with unconjugated TEVGs missing from the work which may offer an insight into the uniqueness of the modified graft.

- The authors could further demonstrate the graft compatibility as claimed using established technique or proteomic work than just the lysosomal markers but test other inflammatory markers as they see significant macrophage infiltration and giant cells.

Minor

- The authors should explain alternative strategies to some of the drawbacks of TEVGs such as structural (basement membrane) and functional (contractility) to successfully develop TEVGs to cater to clinical settings and the disadvantages owing to the lack of contractile function.

- Given the unique protein expression in TEVGs the authors should further elaborate the relevant significance, compatibility, and their role in long term success of the grafts.

- Also, do the authors expect structural formation like the native carotid artery if they extend the time of study?   

- The narrative of the paper seems a little confused, I think authors need to make it clearer that the TEVGs are indeed the way forward or otherwise.

Author Response

(The authors gave the same response as above.)

Reviewer 3 Report

polymers-2017391

 Title: Controlled and synchronised vascular regeneration upon the implantation of iloprost- and cationic amphiphilic drugs-conjugated tissue-engineered vascular grafts into the ovine carotid artery: a proteomics-empowered study

 In this manuscript, the authors combined electrospinning and fused deposition additive manufacturing techniques to fabricate PCL small-diameter tissue-engineered vascular grafts with an anti-aneurysmatic PCL sheath. And the antithrombotic drug iloprost and the antimicrobial cationic amphiphilic drug (CAD) were carried by constructing polypyrrolidone (PVP) hydrogels on the luminal surface of TEVG

 The article focuses on the assessment of atherogenesis, calcification, atherosclerosis and infection, endothelialisation, and differentiation of the inner and marginal layers during in vivo TEVD degradation/vascular regeneration. In particular, proteomic analysis was performed by ultra-performance liquid chromatography-tandem mass spectrometry (UHPLC-MS/MS) to reveal the behaviour of TEVD in vivo at the protein level.

 However, this article lacks the necessary characterisation of materials, as Polymers is a journal primarily concerned with the theory, synthesis, characterisation and application of polymers. Chemical analysis (FTIR, XPS) should be performed after each step of the surface modification. The amount and release behaviour of iloprost and CAD loaded on TEVD also needs to be examined.

 Therefore, I recommend that the manuscript either be reconsidered with a major revision, i.e., supplementing the results of the experiments mentioned above, or published in another journal.

Author Response

(The authors gave the same response as above.)

Round 2

Reviewer 2 Report

The authors have addressed a majority of concerns and significantly improved the quality of the work. I believe the manuscript may be published in current form.

Reviewer 3 Report

The revised manuscript adds sufficient results on material characterisation to demonstrate changes in the chemical structure of the material before and after surface modification and to show the release process of two drugs on board, among other things. Recommended for publication.